# Alteration of Gut Microbiota in Inflammatory Bowel Disease (IBD): Cause or Consequence? IBD Treatment Targeting the Gut Microbiome

**DOI:** 10.3390/pathogens8030126

**Published:** 2019-08-13

**Authors:** Israr Khan, Naeem Ullah, Lajia Zha, Yanrui Bai, Ashiq Khan, Tang Zhao, Tuanjie Che, Chunjiang Zhang

**Affiliations:** 1School of Life Sciences, Lanzhou University, Lanzhou 730000, China; 2Key Laboratory of Cell Activities and Stress Adaptations, Ministry of Education, Lanzhou University, Lanzhou 730000, China; 3Gansu Key Laboratory of Biomonitoring and Bioremediation for Environmental Pollution, Lanzhou University, Lanzhou 730000, China; 4Probiotics and Biological Feed Research Center, Lanzhou University, Lanzhou 730000, China; 5Gansu Key Laboratory of Functional Genomics and Molecular Diagnosis, Lanzhou 730000, China

**Keywords:** inflammatory bowel disease, gut microbiota, pathogenesis, prebiotics, probiotics, synbiotics, fecal microbiota transplantation (FMT)

## Abstract

Inflammatory bowel disease (IBD) is a chronic complex inflammatory gut pathological condition, examples of which include Crohn’s disease (CD) and ulcerative colitis (UC), which is associated with significant morbidity. Although the etiology of IBD is unknown, gut microbiota alteration (dysbiosis) is considered a novel factor involved in the pathogenesis of IBD. The gut microbiota acts as a metabolic organ and contributes to human health by performing various physiological functions; deviation in the gut flora composition is involved in various disease pathologies, including IBD. This review aims to summarize the current knowledge of gut microbiota alteration in IBD and how this contributes to intestinal inflammation, as well as explore the potential role of gut microbiota-based treatment approaches for the prevention and treatment of IBD. The current literature has clearly demonstrated a perturbation of the gut microbiota in IBD patients and mice colitis models, but a clear causal link of cause and effect has not yet been presented. In addition, gut microbiota-based therapeutic approaches have also shown good evidence of their effects in the amelioration of colitis in animal models (mice) and IBD patients, which indicates that gut flora might be a new promising therapeutic target for the treatment of IBD. However, insufficient data and confusing results from previous studies have led to a failure to define a core microbiome associated with IBD and the hidden mechanism of pathogenesis, which suggests that well-designed randomized control trials and mouse models are required for further research. In addition, a better understanding of this ecosystem will also determine the role of prebiotics and probiotics as therapeutic agents in the management of IBD.

## 1. Introduction

Inflammatory bowel disease (IBD) is a chronic inflammatory condition of the gastrointestinal tract (GI), which is characterized by a disrupted mucosa structure, altered gut microbial composition, and systemic biochemical abnormalities [1]. IBD has two major clinical forms, ulcerative colitis (UC) and Crohn’s disease (CD), which are differentiated by the different clinical manifestations of inflammation and intestinal localization [1,2]. Over the past decade, IBD has emerged as a global public health challenge [3], and the frequency of incidence is progressively increasing across the globe [4,5]. The highest incidence rate has been recorded in developed countries, such as North America, Europe, Australia, and New Zealand [6], while recently, IBD has been gradually becoming more common in developing areas such as Asia and South America, and in developing countries such as including Brazil, South Korea, and China [7,8]. The estimated incidence and prevalence of UC and CD in China has increased to 11.6 and 1.4 cases per 100,000 persons per year, respectively [9]. The precise etiology of IBD is unknown, however, the most dominant hypothesis suggests that IBD results from an exaggerated immune response, triggered by environmental factors towards the altered gut microbiota or pathogenic microorganisms in a genetically prone host [10,11,12]. Gut microbiota alteration in IBD pathology is generally accepted; however, it is unclear whether such an alteration is the cause of intestinal inflammation or a consequence of it, and precisely how these bacteria contribute to IBD pathogenesis remains unclear [13,14]. Clarifying these questions would be a milestone in the development of effective therapy for IBD and other gut microbiota-related disorders. IBD treatment options are quite limited due to its complicated and unknown etiology and are focused on disease induction and maintenance in remission rather than on a cure [15]. This lack of an appropriate treatment accounts for the high burden of IBD [16]. Commonly recommended standard therapies for IBD-UC patients are corticosteroids [17], amino-salicylates [18,19], and immunosuppressive agents [20,21]. In addition, biological agents such as infliximab and tacrolimus [22] have been used as part of the regime for UC since 1998 and 2006, respectively [23]. This current treatment is short term and relieves symptomatic complications, but is accompanied by serious side effects, including loss of immune tolerance and drug resistance [24]. The low therapeutic efficacy and high adverse effects of modern therapy are impetus enough to seek an alternative effective therapeutic strategy such as prebiotics, probiotics, and synbiotics, as either complementary or alternative medicines (CAM) to treat IBD and other similar diseases. This review also discusses the alteration of gut microbial flora seen in IBD, the involvement to IBD pathophysiology, and relevant therapeutic approaches in the current literature.

## 2. Alteration of Gut Microbiota (Dysbiosis) and IBD Pathology

The involvement of gut microbiota in IBD pathogenesis has recently been highlighted in the research. The human gut microbial flora is a dynamic and diverse community of commensal bacteria, fungi, and viruses; among them, bacteria, of which there are over 1000 different species, constitute the major part [25,26,27]. More than 90% of healthy human gut bacterial species belong to four major phyla: Bacteroidetes, Firmicutes, Actinobacteria and Proteobacteria [25,26,28,29,30]. However, there is a significant inter-individual microbial diversity difference within these major phylotypes [31]. The gut microbiome has a mutual symbiotic relationship with the human host, in which the host provides a nutrient-rich habitat and residency for the microbiota, while the gut microbiota supports the host through various physiological functions to maintain a healthy state. Under normal physiological conditions, gut microbiota acts as a homeostatic organ involved in the fermentation of complex undigested polysaccharide polymers, production of short-chain fatty acids (SCFAs), synthesis of certain vitamins, energy production, intestinal mucosa integrity, and preclusion of pathogenic microbes [32,33,34,35,36]. In addition, certain symbiotic gut microbiota members have been reported to have distinct and specific effects on the host immune system, and are considered key to immune homeostasis [31,37,38]. The balance of Th17 and Treg cells, which are characterized by pro-inflammatory and anti-inflammatory cytokines, is critical for the host’s intestinal homeostasis and is directly affected by the normal gut microbiota content. Evidence from segmented filamentous bacteria (SFB) in mice has shown a high level of accumulation of pro-inflammatory cytokines of T helper 1 (Th1) and T helper 17 (Th17) [39,40,41]. Bacterial species from the *Clostridia* and *Bacteroides* genera induce anti-inflammatory and T-reg cell responses [42,43,44]. Some other studies have reported that certain bacterial antigenic signals, such as the retinoic acid of *Clostridium* cluster IV and XIVa [45], and polysaccharide A of *Bacteroides fragilis* [46] and *Faecalibacterium prausnitzii* [47], are involved in triggering an immune response and the accumulation of Treg cells. The given mouse model studies suggest that gut microbiota play a role in mucosal immune homeostasis and inflammation, particularly in the Th17/Treg balance, which is considered a dominant factor in induction and inhibition of colonic inflammation [48] (Figure 1).

Hence, the co-existence of the gut microbiota and host reveals the crucial role of the gut microbial flora in host health, and maintenance of the gut microbiota equilibrium is highly important to the host gut and overall systemic physiology. Any change in the steady-state of the gut microbiota structural composition that can alter the microbial equilibrium is termed “dysbiosis” and is associated with a variety of gut pathologies [49] and intestinal inflammation [50]. Therefore, the investigative study of the mammalian gut microbiome is important, particularly focusing on bacteria which are the dominant members of this community [25]. Dysbiosis, a change of the gut microbiota structure composition and/or function, is considered a matter of causation for deterioration of the homeostatic relationship between the microbes and host [28,51,52]. This disruption in the gut microbiota balance ultimately results in an alteration of the gut micro flora-associated functions, including alteration of fermentation products such as carbohydrates, vitamins, and SCFAs [53], and biochemical process alterations, such as immune equilibrium imbalance [54,55,56]. An imbalance of the Th17/Treg correlation and their pro-inflammatory cytokines production is particularly problematic [57,58], as it either causes initiation or propagation of disease pathogenesis, suggesting a link between dysbiosis and disease etiology [59,60,61,62,63,64,65,66,67] (Figure 2). Dysbiosis as a cause of IBD etiopathology has been reported in many studies [60,68,69,70,71,72,73,74]. A significant difference in the gut microbiome of healthy individuals and IBD patients in terms of load and diversity has been confirmed [75,76]. The pattern of dysbiosis most associated with IBD patients is a decrease in commensal bacteria diversity, particularly in Firmicutes and Bacteroides, and a relative increase of bacterial species belonging to *Enterobacteriaceae* [77,78,79,80,81]. Another study has shown the association of five bacterial species, including an increase in *Ruminococcus gnavus* and a decrease in *Bifidobacterium adolescentis*, *Dialister invisus*, *Faecalibacterium prausnitzii,* and an unknown member of *Clostridium cluster XIVa* [82]. Gevers and colleagues demonstrated a correlation between loss of species diversity and the disease activity index of CD patients [83]. More in-depth studies conducted in UC and CD patients have shown a clear reduction in Firmicutes (especially *Clostridium* groups) and an increase in Proteobacteria [28,84,85], alongside a significant decrease of many other beneficial bacterial species from the genera *Bacteriodes*, *Lactobacillus,* and *Eubacterium* [86,87]. According to Schultsz et al. & Manichanh et al., IBD patient gut flora studies have always presented an alteration featuring high load and less diversity [88,89]. Similarly, other studies have reported an overall reduction in the total number of species and a decrease in diversity of the intestinal flora in IBD. Qin et al. suggested that IBD patients harbor only 25% (fewer) of the mucosal microbial genes of healthy individuals [25]. The reduced and altered gut microbiota composition has also been documented in IBD patients’ fecal and mucosa samples [51,89]. A recently published study of 132 IBD patients provided the most comprehensive description to date of host and microbial activities in inflammatory bowel diseases, demonstrating that the gut microbiome and the molecular functional profile in terms of transcriptome and metabolome, and the host immune factors, are central to this disease [90]. Similarly, another recently published study showed a disruption in the taxa of the Lachnospiraceae and Ruminococcaceae families in patients compared to controls, and demonstrated that disturbance in the distinct networks of taxa associations is involved in CD and UC disease development [91]. An altered gut microbiome composition has been identified in several other gastrointestinal diseases, including irritable bowel syndrome, celiac disease, antibiotic-associated diarrhea, functional dyspepsia, tropical enteropathy, and others [92,93,94,95,96]. Although the spectrum of published results has widely recognized the dysbiosis which occurs in IBD patients, the causal role of dysbiosis has not been established yet. In addition, dysbiosis in IBD cases is inconsistent, and the microbiota composition has varied among the surveying studies of IBD patients. This may be partly due to variation in specimen type, sample location, the disease state of subjects, and materials and methods of analysis [97].

Strong evidence for a key causal role of the gut microbiota in IBD development has been provided through mouse models of intestinal inflammation [14]. In general, in most mouse colonic inflammation models of IBD, the pattern of lesions in the colon and distal parts of the ileum are similar to the pattern of inflammatory lesions of human IBD. Importantly, the load of bacteria has been shown to be significantly higher in gut affected areas, in the colon and ileum, but lower in the other parts of the GIT, supporting the association of the gut microbiota with gut inflammatory lesions [28,98]. Similarly, functional evidence for an implication of the gut microbiota in IBD comes from mouse models either treated with antibiotics to decrease the gut microbiota, or from experiments using gnotobiotic mice models in the isolator, where the mice were devoid of any micro-organisms. Generally, in most of the induced intestinal inflammation models, such as the chemically induced dextran sulfate sodium (DSS) and trinitrobenzene sulfonic acid (TNBS) and the genetically induced NEMO−/− mice, interleukin-10 knock-out mice, and HLA-B27 transgenic rats, no inflammation developed in these animals under germ-free gnotobiotic conditions. However, when these animals were exposed to conventional conditions, they developed disease manifestation. This demonstrates the decisive role of gut microbiota in the development of intestinal inflammatory conditions [99,100,101,102,103,104,105]. Further, the decisive role of gut microbiota in colitis was confirmed by the colitogenic potential of colitis-bearing mice gut microbiota transferred to wild-type mice, which induced inflammation [106], as well as the successful resolution of *Clostridium deficile* infection and colitis remission in UC patients who underwent fecal microbiota transplantation (FMT) with samples obtained from healthy individuals [107,108].

Moreover, in order to predict the role of gut microbiota in the intestinal inflammation of IBD, an alternative strategy was applied in many studies in which the gut microbiota was targeted, and the results suggested that altered gut microbiota is key to the induction and maintenance of colon inflammation [109,110]. This notion has been supported by the ameliorative role of prebiotics, probiotics, and antibiotics, which conferred benefits to a certain subset of IBD patients [111,112,113,114]. This knowledge has led to a new domain of research, “modulation of IBD patients’ gut microbiota”, using microbiota targeting therapeutic approaches such as prebiotics, probiotics, and synbiotics to treat IBD [115]. Though great progress has been made in understanding the key role of the intestinal microbiota in intestinal inflammation and IBD pathogenesis [38], precisely how bacteria contribute to disease pathogenesis remains incompletely understood [51,116,117]. Therefore, it is important to clarify how these differentiated altered microorganisms are involved in IBD pathogenesis, and it remains a notable challenge to investigate and reveal their direct or indirect involvement in IBD.

## 3. Specific Individual Bacteria Species or Communities Involved in IBD

The above mentioned studies justify the crucial role of the gut microbiota in the development of gut inflammation; however, it is still unclear whether a specific individual bacterial species/strain or a group of certain bacterial species/strains might be causative of IBD or only contribute in exacerbation of IBD pathogenesis; this is an ongoing debate. Several studies have demonstrated the high prevalence of specific bacterial pathogenic or commensal species/strains in IBD patients, but so far none of these species/strains could be convincingly shown to cause IBD. This question has also been addressed in mouse models infected with enteric pathogens to evaluate the colitogenic potential of an individual specific bacterial species to cause chronic inflammation [118]. For example, *Bacteroides fragilis,* a human commensal bacterial species, causes chronic intestinal inflammation in mice [119]. Similarly, *Klebsiella pneumoniae* and *Proteus mirabilis*, when overabundant in a T-bet-Rag2 mice model, were correlated with the degree of severity of colitis. Moreover, these strains caused colitis in adult Rag2−/− and wild-type mice [120]. Many studies in human IBD patients have reported the commensal bacteria association with the patient’s intestines; for example, an association of adherent/invasive *Escherichia coli* (AIEC), a normal gut flora of the Enterobacteriaceae group, has been found with CD [121]. It has also been reported to be a causative agent in granulomatous colitis of boxer dogs [122]. About 40% of ileal CD patients have overrepresented adherent/invasive *E. coli* (AIEC) compared to healthy controls [123,124]. This assumption has been further strengthened by Mann and Saeed, who demonstrated a high concentration of invasive *E. coli* in more inflamed ileal regions compared to less inflamed or uninflamed regions [125]. AIEC association has only been demonstrated in ileal Crohn’s disease, and has not been observed in the colon in CD, therefore this explanation of AIEC appears to be strictly limited to patients with ileal CD, and does not explain the inflammatory lesion development in the colon with CD. Furthermore, the existence of AIEC in healthy individuals shows that AIEC is a normal gut flora member, and acts as an opportunistic pathogen in a genetically susceptible host in a way that either causes or contributes to disease pathogenesis [125]. Another frequently associated bacterial species in IBD cases is *Mycobacterium avium subsp. Para-tuberculosis* (MAP). MAP is a strict aerobic pathogenic bacterium that has been discussed as a potential triggering factor involved in IBD etiopathogenesis [126]. MAP has been recognized to induce chronic colitis in cattle and other species, including non-human primates and dogs. MAP-induced intestinal inflammation in cattle is termed Johne’s disease, and it almost mimics the clinical and histological features of human CD [127]. Strong evidence of MAP implication in intestinal inflammation comes from many CD patient studies that have demonstrated a higher MAP-DNA level in the mucosal tissue of CD patients compared to healthy controls [128,129,130]. Moreover, engagement of MAP in IBD pathology has been supported by the presence of antibodies and reactive T-cells against MAP in CD patients [131]. More importantly, defective recognition of MAP has been demonstrated in NOD2 variant CD patients, linking MAP to genetic alteration [132,133]. Similarly, the demonstration of MAP persistence in macrophages of mice deficient in the autophagy *ATG16L1* gene proved the role of MAP in intestinal inflammation [134,135]. These explanations indicated that pathogenic bacteria might be causative of chronic intestinal inflammation in genetically susceptible hosts. However, some other studies have demonstrated sharply contrasting results to the aforementioned studies. For example, two different studies have reported a higher level of MAP-DNA in healthy volunteers than in IBD-diseased cases [130,136]. However, the more important controversy against MAP causality in IBD pathogenesis is the fact that no efficient antibiotic treatment effects have been shown in IBD patients [137]. Similarly, another bacterial species, *Clostridium difficile*, has been reported in many IBD clinical cases and demonstrated as either a cause of or a contributing factor to IBD pathogenesis [69,138,139,140]. However, the involvement of *C. difficile* in IBD as a cause has become controversial due to its infection following antibiotic use, providing an important argument that IBD patients might have a higher risk of *C. difficile* infection due to immune-modulatory drugs and antibiotic use or genetic susceptibility. The current thinking is that there is no well-defined evidence to prove that *C. difficile* could be a credible candidate for causing IBD, however, its frequent association with IBD patients more likely shows that *C. difficile* infection is secondary, and contributes to exacerbation in IBD pathogenesis [138,139,140]. These and several other studies (Table 1 and Table 2) have demonstrated dysbiosis and the association of a distinct pathogenic, commensal, or opportunistic bacterial species in IBD patients and in mice colitis models, however, the simple presence of bacteria in IBD patients or mice samples does not provide a clear causal link between the gut microbiota and IBD, because none of these specific pathogens have been confirmed as a cause of IBD as per the Koch’s postulates. To prove this, the Koch’s postulates need to be fulfilled, and to date, no study has met the strict requirements postulated by Robert Koch for a pathogenic microbe in an infection, in order to be able to present IBD is a sort of infectious disease. Therefore, the view of IBD as a kind of infectious disease remains highly controversial.

## 4. Genetics in Dysbiosis and IBD Pathology

Genetic predisposition is one of the etiological factors of IBD, and it is thought that the gut microbial ecology shift in IBD can be described as a consequence of the genetic information of the respective host. The linkage of susceptibility genes with certain bacteria prevalence in IBD has been recently highlighted. Genome-wide association studies (GWAS) have identified more than 160 genetic loci associated with conferring protection from IBD or increased risk to IBD development [172]. The report of a high number of diverse genetic factors involved in IBD pathophysiology supports the hypothesis that IBD is a polygenic disease that can develop due to inefficient handling of the gut microbiota. The defects and variations seen in most of these genes can be attributed to microbial recognition and killing functions, mucosa barrier integrity, and immune regulation [70,173,174]. Therefore, this suggests that IBD might develop in IBD patients due to certain defects of the host in the regulation of commensal bacteria, or even handling of pathogens or regulation of the immune response to these challenges. The NOD2 (nucleotide-binding oligomerization domain containing protein 2), a first gene locus, has been reported as a risk locus for Crohn’s disease (CD) [175,176]. NOD2 is an intracellular pattern recognition receptor which functions to recognize peptidoglycan (muramyl peptide), a certain antigenic part of the bacterial cell wall. NOD2 has two variants; NOD2 heterozygotes carry about a twofold increased risk for CD, which is a relatively lower susceptibility than NOD2 homozygotes which carry about a 20-fold increased risk for CD [175]. These outcomes confirm the involvement of genetic factors that are possibly associated with IBD, however, there are no well-defined results to show how these NOD2 variants influence the risk for CD development. While generally there is a consensus that NOD2 variants have an impact on bacteria handling and regulation, its consequences are controversial and highly debated. For example, normally it is suggested that NOD2 acts as a negative regulator of the Toll-like receptor and prevents excessive activation and stimulation of the NFκB immune signaling pathway [177]. The NOD2 variants might lose this negative repressive role on the Toll-like receptor, leading to dysregulated activation of the NFκB immune signaling pathway, resulting in a high immune response that may be linked to IBD [178]. It is also speculated that CD-associated NOD2 variants might allow intracellular infiltration of bacteria and allow them to escape from immune cell phagocytosis, resulting in a high immune response [179]. Another hypothesis suggests that NOD2 might be involved in antimicrobial peptide (AMP) expression and production in the Paneth cells, while NOD2 variants result in a lower secretion of these proteins into the small intestine, resulting in a bacterial population that promotes an unregulated immune response and causes intestinal inflammation [180]. The impact of the NOD2 gene on microbial regulation has been addressed in an NOD2 gene knock-out mouse model that showed a highly persistent overgrowth of the pathogenic bacterial species *Helicobacter hepaticus* [181]. In addition, enriched colonization of Bacteroidetes and Firmicutes in the NOD2-deficient mice versus control mice indicated a determinant role of NOD2 in microbial community selection and regulation of their composition. The same results have been validated in recent human studies, supporting the presence of NOD2 risk alleles in correlation with a shift of microbial ecology composition in IBD patients’ colons [182,183]. Furthermore, a very elegant experiment in NOD2-deficient mice clearly demonstrated that genetic ablation of NOD2 is a triggering factor of dysbiosis, which is a critical risk component of colitis and colitis-associated carcinogenesis (CAC) in mice [184]. Similarly, another study with an NOD2-deficient mice model reported by Rehman et al. claimed that NOD2 is required for temporal development and composition of the host’s intestinal microbiota [185]. Another gene of interest is *ATG16L1*, which is normally involved in autophagy, however, its deficiency and polymorphism have been described in some subsets of CD patients [186]. *ATG16L1* deficiency has been found in association with a decrease in Paneth cell function, while polymorphism in the *ATG16L1* gene affected the immune pathways which are involved in executing autophagy and in the proper handling of intracellular bacteria [187]. In a human study, patients with CD showed lower expression of the Paneth cell defensin proteins HD5 and HD6, and a reduction in antimicrobial activity [180]. In addition, CD patients have been observed to have a high load of Paneth cell death in inflamed regions [188]. These outcomes strongly suggest the involvement of genetic factors in intestinal inflammation of IBD, and also support the hypothesis that a dysregulated handling of the microbiota might be causative of IBD. Similarly, a mouse immune-compromised model RAG2−/− (mice deficient in the recombination-activating gene (RAG)) colonized with *Helicobacter hepaticus* developed chronic colitis that resembled the inflammation features of human IBD [189], while colitis did not occur in the corresponding RAG immune-competent mice infected with *Helicobacter hepaticus*, implying that the host’s adaptive immunity actively controls the overgrowth of *H. hepaticus*, which is a cause of colitis [190]. Consistent with study [189] outcomes, Kullberg and colleague also reported that immune-competent wild type mice infected with *H. hepaticus* showed T regulatory cells that protected the wild type mice from *H. hepaticus* induced colitis, but not in mice deficient in the RAG−/− gene, suggesting that an induction of mucosal Treg cells by the gut microbiota might play an important role in IBD prevention in normal individuals [191]. Further, the concept that a host with genetic susceptibility fails to deal efficiently with invasive bacteria or pathogens due to defects in their pathogen recognition and handling system was supported by a recent study that showed an induction of colitis in *ATG16L1* gene-deficient mice, while no colitis occurred in control mice that were infected with norovirus [192]. Moreover, the deletion of NEMO (NF-kB essential modulator), a luminal mucosal cell-specific gene involved in NF-kB signal transduction, induced TNF-α driven intestinal apoptosis in mice colon epithelial cells [193]. Similarly, mice deficient in the caspase-8- gene, an intestinal epithelial cell-specific gene, experienced spontaneous ileitis [188]. This spontaneous cell death resulted in defects in the gut barrier function and diminished the intestinal Paneth and goblet cells, implicating defects in the mucosal innate immunity against luminal pathogens/bacteria. This explanation proposes that loss of intestinal epithelial cells and barrier dysfunction may allow translocation of enteric pathogens/commensal bacteria into the gut mucosa’s deep layer, developing an adaptive immune response that might be a driving force behind the etiopathogenesis of inflammatory bowel disease. The aforementioned studies have demonstrated gut microbiota alignment with the presence and absence of genetic alterations, suggesting that possibly IBD could be the outcome of mutations in host genes which are related to the handling of commensal microbiota or pathogen killing functions, and thus have led to an alternative hypothesis that dysbiosis is not causally linked to the pathogenesis of IBD, but rather is a consequence of host genetic alteration. However, more careful analysis is required to define this molecular cross-talk between the gut microbiota and host genome, and their dependencies. The host–microbe interactions in IBD etiopathogenesis are shown in Figure 3.

## 5. Research Progress in IBD Treatment

Due to the worse effects and ineffectiveness of IBD standard chemotherapy, it is necessary to seek an alternative effective therapeutic approach to the treatment of IBD. In this context, scientists have focused on the role of gut microbiota alteration in IBD pathogenesis, by aiming to restore the intestinal microbial flora composition using prebiotics, probiotics, synbiotics, and fecal microbiota transplantation (FMT) as complementary and alternative medicines (CAM) to ameliorate the intestinal inflammation of IBD. These approaches have been assessed in many experimental animal models and clinical trials. The potential effects of these strategies are shown in Figure 2.

### 5.1. Complementary and Alternative Medicine (CAM)

Complementary and alternative medicine (CAM) refers to a broad range of therapeutic and prophylactic approaches including spirit idea therapy, basic biological therapy, body adjustment therapy, and herbal medicine, among others, that provide an alternative to mainstream medical science in patients who have not been cured or who are reluctant to take western medicine. Different from traditional western medicine, CAM is gradually welcomed and accepted by patients because of its mild nature, good curative efficacy, and much lower side effects than western drugs [194]. The trend of CAM, particularly herbal medicine, as a therapeutic agent for the treatment of several chronic conditions including IBD is increasing, and to date, no serious adverse effects have been recorded. The use of CAM in IBD treatment has been reported in many countries, and a reasonable frequency of CAM as a therapeutic agent against IBD has occurred in places such as in France (21.2% of patients) and Germany (51.3%). Many UC patients (59.8%) and CD patients (48.3%) have had experience with CAM, while in Canada, of 2847 IBD patients, 47% had experienced CAM therapies [195]. The most commonly used types of CAM are homeopathy (52.9%) and herbal medicine (43.6%). Among the Canadian population, herbal medicine therapies were the most common, accounting for 41% of CAM users [196], and only 1.6% of the Canadian prior CAM users had experienced potential adverse effects [197]. Herbal therapy exerts therapeutic effects in the management of IBD by various potential mechanisms, including restoration of the gut microbiota, immune system regulation, inhibition of leukotriene B4, antioxidant activity, inhibition of nuclear factor-kappa B (NF-jB), and antiplatelet activity [198].

### 5.2. Traditional Chinese Herbal Medicine (TCMs) and IBD

Throughout human history, the role of traditional Chinese herbal medicine (TCMs) as CAM in disease control and cure has been indispensable [199,200]. Chinese HMs with natural raw materials have low adverse and high curative effects compared to western drugs, and have become a hot spot in IBD drug research. The effects of traditional Chinese medicine in the context of IBD have been reported using a variety of traditional TCMs decoction formulas, such as the Wumei pill decoction, Chaihu peony soup, and Pulsatilla decoction, which have shown significant efficacy in the alleviation of IBD pathogenesis by regulating the gut microbial ecology [201,202]. Although the polysaccharides of Chinese HMs are indigestible, and have been less reported in the restoration of the intestinal flora in IBD patients, their pharmacological effects such as immunosuppressive advantage and coexistence with gut microbes have made them excellent prebiotics in IBD adjuvant therapy [203]. Astragalus polysaccharide administration has shown a better effect in TNBS-induced colitis alleviation by prevention of bacterial translocation, significantly increasing beneficial bacteria such as *Lactobacillus* and *Bifidobacteria*, and decreasing *Enterobacteriaceae* and enterococci [204]. Similarly, Purslane polysaccharides have also enhanced *Bifidobacterium* and *Lactobacillus*, while reducing the peripheral blood endotoxin content in a DSS-induced ulcerative colitis mice model [205,206]. Chinese HM polysaccharides have also shown a regulatory effect on Th1 and Th2 cells, and are expected to play a significant role in immune disorder treatments. The Huangqin-Tang decoction (HQT) restored the Th1/Th17 drift in TNBS-induced colitis by reduction of pro-inflammatory cytokines and their transcription factors, while improving the intestinal protective immunity by enhancing the Th2/Treg-associated cytokines. [207]. Similarly, Astragalus polysaccharides reduced the pro-inflammatory cytokines IL-17A and IL-25, and increased the anti-inflammatory cytokines IL-35 and IL-10, as well as regulating the imbalance of Th17/Treg cytokines in an asthmatic rat model. Ginseng origin acid polysaccharides have shown downregulation of IL-1 and IL-17 expression, and upregulation of the FOXP3 gene and Treg cells, in animal colitis model [208].

### 5.3. Herbal Medicine as Prebiotics in IBD Treatment

Prebiotics are indigestible food ingredients, usually carbohydrates, which can promote the growth and activity of selected beneficial gut microbes and provide benefits to the host’s well-being by the amelioration of enteric dysfunction [113]. Prebiotics primarily act on certain bacterial populations and enhance their growth, resulting in the restoration of the normal composition of altered microbial flora [112]. Within the colon microenvironment, prebiotics are metabolized by anaerobic gut microbiota, producing short-chain fatty acids (SCFA) and gas (CO2 and H2) as fermentation products, which reduces the colon pH so as to favor the growth of *Bifidobacteria, Lactobacilli*, and non-pathogenic *E. coli,* while reducing *Bacteroidaceae* and other potentially pathogenic bacteria [115,209]. Fermentation products such as acetic, propionic, and butyric acids of carbohydrates are involved in the activation and regulation of multiple colon-specific and systemic pathways [210]. Acetate is a cell energy source, and propionic acid is involved in cholesterol manufacturing. However, butyrate is of great importance as it is involved in colonocyte metabolism [211]. Butyrate has also been implicated in the remittance of inflammation by imparting anti-inflammatory actions, inhibition of IL-12, downregulation of TNF-α-related cytokines, and upregulation of IL-10 in human monocytes [212], and also possibly by reduction or inhibition of the nuclear translocation of NF-kB [213,214,215]. Butyrate enemas have also been used in UC patients with success, but the need for recurrent administration has limited its use [216]. Prebiotics are considered a unique remedy for inflammatory disease, however, very limited clinical studies have addressed prebiotic use as a therapeutic approach in IBD. The most investigated prebiotic is germinated barley foodstuff (GBF), which is rich in glutamine and hemicellulose as its main components [217] and has shown a significantly effective role in patients with mild-to-moderate UC [218]. GBF improved the Disease Activity Index (DAI) by elevating the growth of protective bacteria like *Bifidobacterium* and *Eubacterium* [218,219] and the fecal butyrate level [220], and by lowering the serum level of CRP [221] and pro-inflammatory cytokines IL-6 and IL-8, along with the suppression of mast cells in UC patients [222,223]. Similarly, other studies have investigated the efficacy of fructooligosaccharides (FOS) and have shown a significant increase in the butyrate level [218], and an increase of beneficial bacteria in CD patients and a TNBS-induced colitis model of rats [224,225]. Oligo-fructose-enriched inulin has shown a decrease in fecal calprotectin in UC patients, indicating a reduction in inflammation [226]. Inulin supplementation has also produced an increased level of butyrate [227,228], alteration of microflora [229], and a decrease in endoscopic histopathology [230]. The available studies show some degree of prebiotic efficacy in IBD treatment, but display inconsistency and variation in outcomes, perhaps due to differences in doses, sample size, and study protocol. The guidelines are very cautious in defining the exact role of prebiotics in IBD and do not allow the use of prebiotics in either UC or CD [231]. Therefore, scientists interested in the use of a mutually supportive combination of pre and probiotics in the form of synbiotics have begun to pursue a solution [116], however more in-depth studies are required to develop a universally effective study protocol for the evaluation of prebiotics in IBD clinical and laboratory experimental trials.

### 5.4. Probiotics

When the gut microbiota variation was first acknowledged in health issues, especially in gastrointestinal disorders including IBD, probiotics were considered a milestone approach to treatment and health restoration. Probiotics are living microorganisms with certain potential activities that can confer health benefits to the host when administered in adequate amounts [112]. The probiotics from dairy products have shown several mechanisms of potential action related to IBD prevention and control, including an antimicrobial effect, suppression of pathogenic bacteria, immune-modulation, enhancement of anti-inflammatory responses, and improvement of the intestinal barrier activity in IBD patients [75,109,232,233,234,235,236]. Various probiotics strains have been isolated from food culturing, especially dairy products, including *Lactobacillus* species, *Bifidobacterium* species, *E. coli Nissle* 1917 (a non-pathogenic *E. coli* strain), *Saccharomyces boulardii*, *C. butyricum*, VSL#3 species, and *Lactococcus lactis* [115]. The most commonly evaluated single probiotic strains that have been reported to be effective and as safe as mesalazine for the maintenance of remission in active UC patients are *Lactobacillus GG* [237,238]*, Escherichia coli* Nissle 1917 [239,240], *Bifidobacteria* [241], and the yeast strain *Saccharomyces boulardii* [242]. Highly supportive data for probiotic use in IBD comes from studies on VSL#3, a commercial probiotic supplement that is highly concentrated (450 billion bacteria/sachet), containing eight different probiotic strains, which has shown benefits relating to clinical remission in active UC patients [243,244,245]. A meta-analysis study published recently reported that the VSL#3 cocktail, given in combination with IBD/UC standard therapy, showed significantly higher curative effects in clinical remission compared to standard therapy alone in UC patients [246]. A few studies published on this topic have also reported that VSL#3 induced an increase in the concentration of *Bifidobacterium, Lactobacillus* and *Streptococcus salivarius* in the gut, however, its effect returned to the basal levels 15 days after treatment [243]. Probiotic effects have also been reported in studies using probiotic naked DNA [247,248,249]. Although the beneficial effects of probiotics have been established in treating various gut pathologies, there is insufficient data supporting the impact of probiotics on the host’s microbiome restoration and health recovery. Thus, further study is needed on the evaluation of probiotic effects in the amelioration of intestinal disorders, particularly in IBD, particularly studies focusing on experimental subjects in laboratory animal models and in clinical settings.

### 5.5. Synbiotics

Synbiotics are combined pro- and prebiotics that can confer a mutual synergistic beneficial effect on host health [112,241]. Pro- and prebiotics in combination are considered a novel approach and there is currently a promising opportunity to evaluate their efficacy and potential use in IBD using clinical settings and animal models. However, a few published studies already exist supporting the use of symbiotic supplementation in IBD. The most frequent and commonly used symbiotic formulae include *Lactobacillus GG* and inulin, *Bifidobacteria* and fructooligosaccharides (FOS), and *Bifidobacteria* and lactobacilli with FOS or inulin. Roller *et al.,* used a mixture of oligofructose-enriched inulin, *Lactobacillus,* and *Bifidobacterium,* and reported the suppression of colonic carcinogenesis and improvement of IgA and IL-10 secretion in the colon [250]. The combination of *Bifidobacterium longum* and inulin-oligofructose, and *B. longum* and psyllium, both used in a randomized control trial as a supplement to conventional therapy in UC patients, showed a good synergistic effect and improved the disease clinical activity index compared to a probiotic or prebiotic alone [251,252]. Similarly, a *B. longum*, inulin, and oligofructose mixture used in a randomized double-blind controlled trial with UC patients demonstrated a reduction in endoscopic inflammation, and TNF-a and IL-1b levels [253]. The *B. breve* Yakult strain and galactooligosaccharides mixture showed a significant anti-inflammatory effect in mild-to-moderate UC patients [254]. Similarly, short bowel syndrome was relieved upon administration of a supplement containing *Bi. breve*, *Lactobacillus casei,* and galactooligosaccharides [255]. Although synbiotics confer more health benefits than pro- or prebiotics alone, the research community is unable to draw definitive conclusions because of the variation in their benefits, perhaps based on the types and doses of pre- and probiotics in different combinations. However, the approach of synbiotics is a new area of research to investigate their effects on the pathogenic mechanisms of gut inflammation, perhaps to find a potential treatment for IBD and other gut-related disorders [115]. Therefore, more human and animal studies are needed to collect convincing data and provide a better understanding of their direct effects on health, particularly in IBD.

### 5.6. Fecal Microbiota Transplantation (FMT)

Fecal microbiota transplantation (FMT) is an emerging technique to treat patients with dysbiosis by the restoration of their abnormal gut microbiota composition via transplantation of normal fecal microbiota obtained from healthy donors. FMT as a potential treatment tool for a variety of gut and other metabolic disorders has been investigated in many animal studies, and has found particular success in patients suffering from severe relapsing *Clostridium difficile* infection [256]. A randomized control study that compared FMT with antibiotics to treat patients for recurrent *Clostridium difficile* infection found quite striking differences [257]. In the FMT group, *C. difficile* associated diarrhea was resolved in 13 out of 16 patients (81%), while in the antibiotics group, only four recovered out of 13 patients (31%). A plethora of findings suggest a crucial role of the gut microbiota in IBD. FMT has gained much attention recently and has been highlighted as a new therapeutic approach in IBD. The first effort of FMT to treat UC was made in 1989 [258]. The largest study conducted on this topic, which covered a wide range of CD patients, was published recently, and reported that FMT is one of the safest, most efficient, and most effective treatments for refractory CD [259]. Similarly, a randomized trial including approximately 70 UC patients conducted by Moayyedi and colleagues demonstrated that FMT was safe and induced disease remission in more patients compared to placebo [108]. Moreover, a recently published systematic review reported that among 18 UC patients with no *C. difficile* infection, 13 UC patients achieved no colitis disease when treated with FMT [260]. A phase 1 trial study conducted on pediatric UC patients reported that FMT was quite safe, with a high rate of clinical response (79%) within one week, and no serious side effects were observed [261]. However, Ott et al. recommended a sterile fecal filtrate transfer (FFT) instead of intact microorganisms as an FMT suspension to avoid any risk associated with donor fecal microbiota transplantation [107]. Unfortunately, some other studies have shown contrasting results, in which FMT failed to ameliorate disease pathology or restore dysbiosis. For example, a randomized UC trial reported that no significant differences were found in recipient patients who were treated with fecal transplants obtained from healthy donors [262]. Moreover, two recently published prospective studies in which FMT was used for adult UC patients failed to achieve clinical remission after FMT, though in most of the patients’, significant structural changes were observed in the gut microbiota [263,264]. One patient from study [263], who experienced clinical betterment, showed a very similar gut microbiota composition to the microbiota of the healthy donor for a longer period of time, suggesting that successful transplantation and colonization of microbiota in the recipient patient correlated with the clinical improvement of the patient. However, another study did not show these findings, despite both studies reporting a temporary shift in the gut microbiota. This raises the possibility that periodic and repeated FMT is necessary to maintain the donor gut microbiota composition in the recipient patient to cure dysbiosis. A systematic meta-analysis study published recently concluded that the reported results of FMT as a treatment approach in IBD are not consistent, and sometimes are quite difficult to interpret [265]. Currently, the number of controlled clinical trials testing FMT in IBD are very limited, and most of these published results have either come from case reports or small cohorts of patients. In summary, whether FMT is beneficial in IBD is mostly unclear. The way to move forward on this topic is to identify the core microbiome related to disease onset or indicative of the disease severity, as this will potentially enhance the chances of achieving clinical remission.

## 6. Conclusions

IBD is a global emerging challenge in the health sector. The presently used chemotherapies and clinical management of IBD are not generally successful due to conflicting etiology. Among the potential causative agents for IBD, the gut microbiota has been suggested as a novel factor. Various research studies have suggested that the alteration of the gut microbiota composition (dysbiosis) is the key stakeholder associated with IBD pathology. Unfortunately, the specific core composition and metabolic biomarkers of the gut microbiome that are suspected to be involved in the onset of IBD pathogenesis remain unclear. Therefore, the question of whether the alterations in gut microbiota composition are a cause or consequence of IBD is still under debate. However, an obvious role of the gut microbiota in host physiology has been demonstrated by the link between gut microbiota dysfunction and host metabolic disorders. Thus, gut microbiota restoration is presumed to be an effective treatment approach. The therapeutic supplementation of probiotics, prebiotics, and synbiotics, and fecal microbiota transplantation (FMT), all seek to alter the gut microbial community and restore a healthy composition in order to alleviate the pathophysiology associated with IBD. We summarized in this review the association of the human gut microbiome with intestinal inflammation, demonstrated the gut microbiome–host interactions and their effect on gut mucosa homeostasis, and we also discussed gut microbiota targeting therapeutic approaches in IBD management with the help of animal models and IBD clinical trials. Our analysis suggests a consistent dysbiosis association with IBD patients, but we have not shown a clear causal link in IBD pathogenesis. However, the gut microbiota has been shown to have a great impact on disease phenotypes and activity in mice colitis models. The gut microbiota-based therapeutic approaches have also demonstrated effects in amelioration of colitis in IBD animal models (mice) to some degree, but less so in human clinical trials. Small amounts and conflicting data necessitate more animal models and clinical trials to be conducted in order to better understand this ecosystem and clarify the concept of dysbiosis in IBD as a causal factor, as well as to identify the potential bacterial strains or core microbiome involved in IBD. In addition, a better understanding of this ecosystem would also determine which bacterial strain or prebiotic would be the ideal treatment for a given bowel disorder. Therefore, more in-depth studies on the subject of the cause and management of IBD are required.

## 7. Future Outlook

There is a clear consensus regarding the involvement of gut microbiota in IBD development; however, the changes in the gut microbial ecology and involvement of specific bacterial species are under debate. Currently, the available studies are retrospective, examining the microbiota after disease onset in IBD patients. To investigate an evidence-based real cause and effect relationship between the gut microbiota and IBD, prospective studies must be undertaken. Such investigative studies will have to be backed by experiments involving the colonization of wild type, germ-free, and genetically modified mice with an individual bacterial species or with a combination of bacteria, in order to identify the exact causal bacterial strain or core microbiome and clarify the fate of the gut microbiota in IBD.

## Figures and Tables

**Figure 1 pathogens-08-00126-f001:**
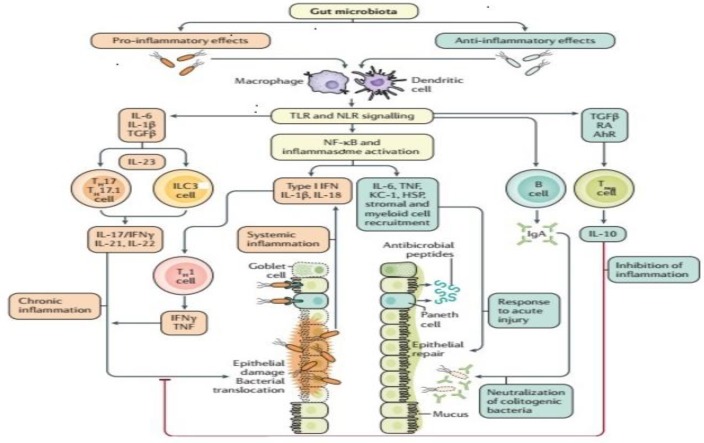
Gut microbiota in induction and inhibition of intestinal inflammation.

**Figure 2 pathogens-08-00126-f002:**
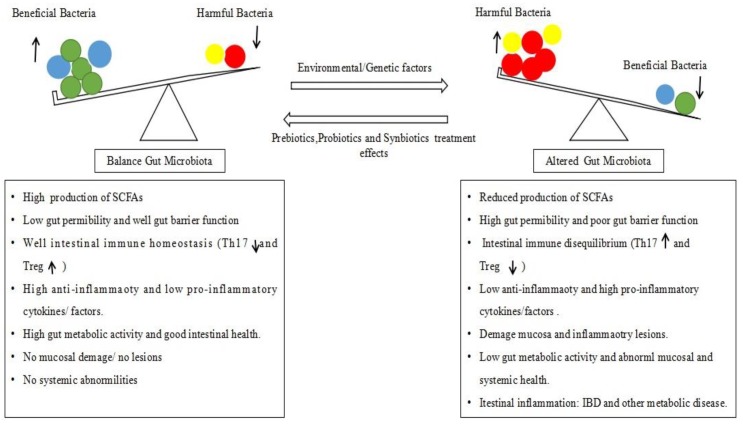
The gut microbiota of the healthy individual (left), the gut microbiota of the inflammatory bowel disease (IBD) patient (right), and pre, pro, and synbiotic effects.

**Figure 3 pathogens-08-00126-f003:**
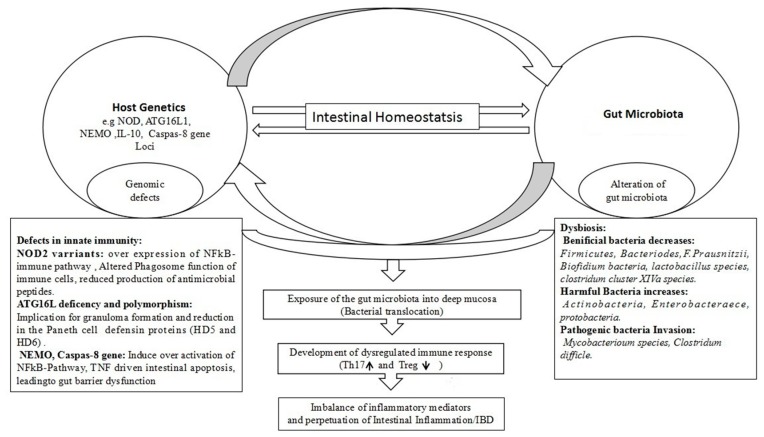
The host–microbe interactions in IBD etiopathogenesis.

**Table 1 pathogens-08-00126-t001:** Dysbiosis of the gut microbiome in IBD (Crohn’s disease (CD) and ulcerative colitis (UC)) patients.

Model	Bacteria	Comments	Ref.
Human	*Firmicutes (F. prausnitzii)*	Decrease	[141]
Human	*Bifidobacterium (Bifidobacterium adolescentis, D. invisus R. gnavus*	Decrease Increase	[82]
Human	*Clostridium* clusters (IV and IXV)	Decrease	[142]
Human	*Enterobacteriaceae (E. coli)*	Increase	[143]
Human	*Lactobacillus*	Decrease	[144]
Human	*Bacteroidetes,* Enterobacteriaceae Firmicutes	Increase Decrease	[71]
Human	*Helicobacter species (H. pylori)*	Increase	[145]
Human	Mycobacterial species (MAP)	Increase	[146]
Human	*Proteobacteria (E. coli* and non-jejuni Campylobacter)	Increase	[147,148]
Human	*Firmicutes and Bacteroidetes, Lachnospiracea* *Proteobacteria, Actinobacteria, Bacillus*	Decrease Increase	[28]
Human	*Staphylococcaceae, Strepotococcaceae, P. maltophilia, Klebsiella, Salmonella*	Increase	[149]
Human	*Faecalibacterium prausnitzii Escherichia coli, Fusobacterium*	Decrese Increase	[150]
Human	*Firmicutes, Faecalibacterium prausnitzii, Bifidobacteria*	Decrease	[151]
Human	*Bacteroidetes, Bacteroides, Flavobacterium, and Oscillospira* *Proteobacteria, Verrucomicrobia, Fusobacteria, Escherichia, Faecalibacterium, Streptococcus*	Decrease	[152]
Human	*Clostridium leptum, Faecalibacterium prausnitzi. Bacteroides spp*	Decrease Increase	[153]
Human	*Clostridia spp., Bacteroides Bacteroides fragilis, proteobacteria*	Decrease Increase	[154]
Human	*enterobactreiace*	Increase	[155]
Human	*Firmicutes: Roseburia, Phascolarctobacterium Enterobacteriaceae: Escherichia/Shigella, Ruminococcus gnavus*	Decrease	[156]
Human	*Firmicutes, Bacteroides species, Eubacterium species, Lactobacillus species Proteobacteria Enterobacteriaceae,*	Decrease Increase	[157]
Human	*Faecalibacterium and Roseburia Enterobacteriaceae and Ruminococcus gnavus*	Decrease Increase	[52]
Human	*Bacteroides, Lactobacillus, Ruminococcus, and Bifidobacterium Peptostreptococcus, Campylobacter, Methanobrevibacter*	Decrease Increase	[158]
Human	*Prevotella copri, Faecalibacterium prauznitzii Enterobacteriaceae*	Decrease Increase	[159]
Human	*Pseudomonas*	Increase	[160]
Human	*Roseburia hominis, Faecalibacterium prausnitzii*	Decrease	[161]
Human	*Proteobacteria, Bacteroidetes, Clostridia*	Unchanged	[143]
Human	*Bifidobacterium, E. coli Firmicutes, C. Coccoides, C. leptum,*	Increase	[162]
Human	*Bacteroidetes Firmicutes, Enterobacteriaceae*	Decrease Increase	[163]
Human	*Firmicutes, F. prausnitzii Bacteroidetes, Proteobacteria*	Decrease Increase	[71,164]
Human	*Firmicutes, Lachnospiraceae, Bacteroidetes, Actinobacteria, Bifidobacteriaceae, Proteobacteria*	Decrease Increase	[83]

**Table 2 pathogens-08-00126-t002:** Dysbiosis of the gut microbiome in mice colitis models.

Model	Bacteria	Comments	Ref.
DSS-colitis	*Bacteroides distasonis, Clostridium ramosum, Akkermansia muciniphila, Enterobacteriaceae*	Increase	[165,166]
TNBS colitis	*Enterobacteriaceae, Bacteroides*	Increase	[167]
T-bet−/−1, Rag2−/− mice*	*Mucispirillum, Desulfovibrio, and Helicobacteraceae*	Increase	[168]
Gonobiotic mice	change in species diversity	Species diversity decrease	[169]
Colitis in IL-10−/− mice*	*Enterobacteriaceae* and adherent-invasive *E. coli*	Increased	[106,170]
Colitis inApc468/IL-10−/− mice*	*Bacteroides* and *Porphyromonas* genera	Increased	[171]

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
