# Peer review of "Alteration of Gut Microbiota in Inflammatory Bowel Disease (IBD): Cause or Consequence? IBD Treatment Targeting the Gut Microbiome"

_pathogens, 2019, doi:10.3390/pathogens8030126_

Round 1
Reviewer 1 Report
Khan et al. nicely revised the current status of IBD studies in the context of microbiota and therapeutic approaches. The context is well designed and subsections of manuscript describe each context adequately. One important thing that is missing is the studies from 2019. There are at least two critical studies that are summarizing most of the microbiota profile that they mentioned in this review. Therefore, I'd like to ask author kindly to check the further literature from 2019 and add at least these two critical manuscripts into their review. These manuscripts provides intellectual inputs for their review.
1 -https://www.nature.com/articles/s41591-018-0308-z
2 - https://www.nature.com/articles/s41586-019-1237-9
Besides that, there are numerous typos in the manuscript. It is important that authors go through the text carefully and revise the manuscript.
Some of those typos and sentences that need to be corrected are as the following:
- There are some bold sentences or group of words should be converted non-bold state.
- Figures' quality are relatively bad and authors should provide a better resolution.
Line 34 - "modles" should be "models"
Line 35 - 37 - Sentence needs to be revised. Grammatically wrong.
Line 38 - Less and conflicting results? Not clear!
Line 48 - Chron’s disease need to be corrected into Crohn’s disease
Line 113 - (SCFAs) is already introduced in Line 86. No need to repeat the meaning of abbreviation.
Line 120 - 124 - The authors show the microbial differences in IBD patients compared to control but they should also mentioned differences between CD vs Control or UC vs Control.
Line 186-202 - species is missing "s" at the end.
Line 192 - enterobacteriacae should be Enterobacteriacae
Line 231 - 234 - The sentence needs to be revised. It is repetative of the same statement.
Line 234 - "Needs" to be
Line 271 - in to should be "into"
Line 285 - sub sets should be subsets
First letter of Enterobacteriacae and Bifidobacteria should be capitalized. It was observed in couple of places that this is not the case.
Author Response
Please, find a point-by-point response to the reviewer-1 report , as an attachment named Response to reviewer-1.

Reviewer 2 Report
This review article is scientifically designed, and the presentation is also overall and elaborated form data collection to literature review, making the conclusion very convincing.The topic of the paper is very relevant. However, some minor revisions are needed.
Manuscript revision
Line 25. Replace Crohon’s with Crohn’s
Line 26. Review the grammatical in “Although the etiology of IBD is unknown, however the gut microbiota alteration 26 (dysibiosis) considered as a novel factor to be involved in IBD pathogenesis”.
Line 38
Replace Less with less
Line 66
Replace [18, 19] with [18,19]
Lines 117 and 204
Replace etio-pathology with etiopathology
Line 127
Replace [28,84, 85] with [28,84,85]
Line 129
Relocate [88, 89]
Line 136
Replace entero-pathy with enteropathy
Lines 168 and 171
Replace pre-biotics with prebiotics
Line 204
Replace etio-pathogenesis with etiopathogenesis
Line 511
In the Conclusion section, the wording should be improved
Line 536
Replace necessary with necessary.
Line 553
Replace colleague with colleagues.
Line 564
Replace “Nature reviews Gastroenterology & hepatology” with “Nature Reviews Gastroenterology & Hepatology”
Line 567
Replace “The American journal of gastroenterology” with “The American Journal of Gastroenterology”. There are similar problems in other references.
Line 568
Replace 647 with 647-654
Line 585
Replace 13 with 13-27
Line 592
Replace 41 with 41-50
Line 594
Replace 64 with 64-69
Line 610
Replace 3 with 3-4
Line 615
Replace 185 with 185-210
Line 617
Replace “A human gut microbial gene catalogue established by metagenomic sequencing. nature, 464(7285), 59” with “A human gut microbial gene catalogue established by metagenomic sequencing. Nature, 464(7285), 59-65”
Line 633
Replace “Sci Trans Med 1, 614” with “Sci Trans Med 1(6), 6-14”
Line 640
Replace 69 with 69-75
Line 642
Delete 33 at the end of the line
Line 644
Replace 85 with 85-93
Line 649
Replace 451 with 451-455
Line 651
Replace 159 with 159-169
Line 677
Replace 573 with 573-584
Line 689
Replace 18 with 18-26
Line 695
Replace 577 with 577-591
Line 706
Replace 691 with 691-701
Line 711
Replace 321 with 321-325
Line 713
Replace 541 with 541-546
Line 716
Delete 52 at the end of the line
Line 717
Replace 800 with 800-812
Line 754
Replace 2143-5 with 2143-2145
Line 789
Replace 2219 with 2219-2241
Line 794
Replace
Matsuoka, K., & Kanai, T. (2015, January). The gut microbiota and inflammatory bowel disease. In Seminars in immunopathology (Vol. 37, No. 1, pp. 47-55). Springer Berlin Heidelberg
with
Matsuoka, K., Kanai, T. (2015). The gut microbiota and inflammatory bowel disease. In Seminars in Immunopathology, 37(1), 47-55
Line 848
Replace 564 with 564-567
Line 854
Replace 1192 with 1192-1210
Line 883
Replace 295 with 295–306
Line 901
Replace 4892 with 4892–4904
Line 925
Replace 1259 with 1259–1267
Line 984
Replace 2619 with 2619–2629
Line 987
Replace 1403 with 1403-1417
Line 1000
Replace 119 with 119-124
Line 1006
Replace 710 with 710-712
Line 1009
Replace 599 with 599-603
Line 1011
Replace 603 with 603-606
Line 1019
Replace 800 with 800-809
Line 1034
Replace 123(2) with 123(2), 700-711
Line 1037
Replace 207 with 207-211
Line 1042
Replace 259 with 259-263
Line 1044
Replace 335 with 335-339
Line 1045
Replace
Ward, J. M., Anver, M. R., Haines, D. C., Melhorn, J. M., Gorelick, P., Yan, L., & Fox, J. G. (1996). Imflammatory large bowel disease in immunodeficient mice naturally infected with. Helicobacter hepaticus. Lab. Anim. Sci, 46, 15.
With
Ward, J. M., Anver, M. R., Haines, D. C., Melhorn, J. M., Gorelick, P., Yan, L., & Fox, J. G. (1996). Imflammatory large bowel disease in immunodeficient mice naturally infected with Helicobacter hepaticus. Lab. Anim. Sci, 46(1), 15-20.
Line 1056
Replace 557 with 557-561
Line 1070
Replace 1563 with 1563-1568
Line 1206
Replace 2218 with 2218-2227
Line 1257
Replace 1620 with 1620-1630
Line 1263
Replace 5359 with 5359-5371
Author Response
Please, find a point-by-point response to the reviewer-2 report , as an attachment named Response to reviewer-2.
